# Alternative Non-Drug Treatment Options of the Most Neglected Parasitic Disease Cutaneous Leishmaniasis: A Narrative Review

**DOI:** 10.3390/tropicalmed8050275

**Published:** 2023-05-14

**Authors:** Mohamed A. A. Orabi, Ahmed A. Lahiq, Ahmed Abdullah Al Awadh, Mohammed Merae Alshahrani, Basel A. Abdel-Wahab, El-Shaymaa Abdel-Sattar

**Affiliations:** 1Department of Pharmacognosy, College of Pharmacy, Najran University, P.O. Box 1988, Najran 66454, Saudi Arabia; 2Department of Pharmaceutics, College of Pharmacy, Najran University, P.O. Box 1988, Najran 66454, Saudi Arabia; aalahiq@nu.edu.sa; 3Department of Clinical Laboratory Sciences, Faculty of Applied Medical Sciences, Najran University, P.O. Box 1988, Najran 66454, Saudi Arabia; aaalawadh@nu.edu.sa (A.A.A.A.); mmalshahrani@nu.edu.sa (M.M.A.); 4Department of Pharmacology, College of Pharmacy, Najran University, P.O. Box 1988, Najran 66454, Saudi Arabia; babdelnaem@nu.edu.sa; 5Department of Medical Microbiology and Immunology, Faculty of Pharmacy, South Valley University, Qena 83523, Egypt; elshaymaa_a_m@svu.edu.eg

**Keywords:** cutaneous leishmaniasis, cryotherapy, photodynamic therapy, thermotherapy, laser therapy, leech therapy, cauterization therapy

## Abstract

With more than 12 million cases worldwide, leishmaniasis is one of the top 10 neglected tropical diseases. According to the WHO, there are approximately 2 million new cases each year in foci in around 90 countries, of which 1.5 million are cutaneous leishmaniasis (CL). Cutaneous leishmaniasis (CL) is a complex cutaneous condition that is caused by a variety of *Leishmania* species, including *L.* (*Leishmania*) *major*, *L.* (*L*) *tropica, L.* (*L*) *aethiopica*, *L.* (*L*) *mexicana, L.* (*Viannia*) *braziliensis*, and *L.* (*L*) *amazonensis*. The disease imposes a significant burden on those who are affected since it typically results in disfiguring scars and extreme social stigma. There are no vaccines or preventive treatments available, and chemotherapeutic medications, including antimonials, amphotericin B, miltefosine, paromomycin, pentamidine, and antifungal medications, have a high price tag, a significant risk of developing drug resistance, and a variety of systemic toxicities. To work around these limitations, researchers are continuously looking for brand-new medications and other forms of therapy. To avoid toxicity with systemic medication use, high cure rates have been observed using local therapy techniques such as cryotherapy, photodynamic therapy, and thermotherapy, in addition to some forms of traditional therapies, including leech and cauterization therapies. These CL therapeutic strategies are emphasized and assessed in this review to help with the process of locating the appropriate species-specific medicines with fewer side effects, lower costs, and elevated cure rates.

## 1. Introduction

Leishmaniasis comprises a complex of vector-borne diseases caused by more than 20 species of the *Leishmania* parasite [1]. According to the infective species, the disease severity may range from localized skin ulcers to lethal systemic disease. Leishmaniasis ranks fourth in morbidity and second in mortality among all tropical diseases [2]. Except for Australia and Antarctica, it affects the most vulnerable populations, including those in emerging nations, residents of focal areas in tropical, subtropical, and southern European countries, and the majority of the world’s poorest nations [3]. Because of its close connection to poverty and the resulting limited funding assigned for diagnosis, treatment, and control, leishmaniasis is one of the most neglected tropical diseases. Out of the 2 million new cases of leishmaniasis that are diagnosed each year, 1.5 million are cutaneous leishmaniasis (CL) [4]. Since ancient times, cutaneous leishmania has been given a variety of common names based on the characteristics of the skin lesion, how it is described in different countries, the name of the endemic area, or the occupation of the affected individuals. These names include Aleppo boil, Aleppo button, Aleppo evil, Baghdad boil, Biskra button and Biskra nodule, Calcutta ulcer, chiclero ulcer, Delhi boil, Jericho button, Kandahar sore, Lahore sore, Oriental button, and Oriental sore, Pian bois, and Uta [5]. How serious the illness is and how to treat it appears to depend on the parasite type, host, endemic region, socioeconomic status, and availability of medical care [6]. Chemotherapeutic drugs for the treatment of CL, including antimonials, amphotericin B, miltefosine, paromomycin, pentamidine, and antifungal drugs, have different systemic toxicity, higher cost, and drug resistance (Table 1) [7]. To get over these limitations, more modern leishmaniasis medication-treatment strategies have been created, such as immunotherapeutic and immunochemotherapeutic medicines and nanotechnology-based drug delivery techniques; however, according to reports, efforts need to be directed toward the rational investment in new therapies and treatment strategies against the disease [8]. Over the past years, local treatments, including local therapy techniques such as cryotherapy, photodynamic therapy, and thermotherapy, and some forms of traditional therapy, including leech therapy and cauterization therapy, have been established as alternatives to systemic drug administration without experiencing toxicity [8].

In this review, we have highlighted the techniques and underlying mechanisms of these non-drug approaches to treating CL. Additionally, we have gathered and tabulated the findings from numerous studies for each therapeutic strategy, which may enable physicians to choose the most efficient treatment strategy with the fewest side effects.

**Table 1 tropicalmed-08-00275-t001:** A comparative list of the common anti-*Leishmania* chemotherapeutic drugs, their modes of action, and associated toxicities.

Drug	Mode of Usage	Mode of Action	Mild to Moderate Adverse Effects	Toxicities	
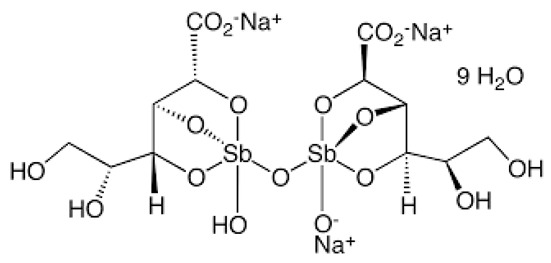 **Sodium stibogluconate**	Intralesional (IL) for CL, parenteral for visceral leishmaniasis (VL)	Inhibit the parasite’s glycolysis and fatty acids *β*-oxidation	Abdominal pain, nausea, and erythema	Hepatic, pancreatic, renal, and cardiotoxicities, thrombocytopenia, or leukopenia	[9]
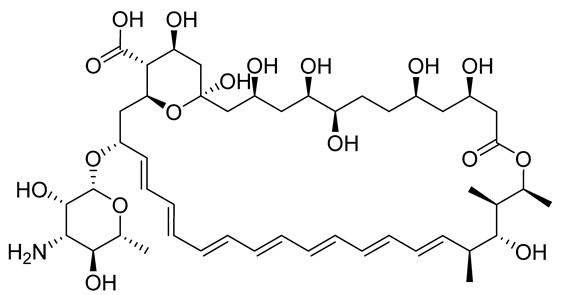 **Amphotericin B**	Liposomal formulations, Deoxycholate formulations	Binds the membrane sterols of the parasite and alters its permeability to K^+^ and Mg^2+^, selectively	Fever, nausea, hypokalaemia, and anorexia	Renal failure, leukopenia, cardiopathy, and hypokalaemia	[10]
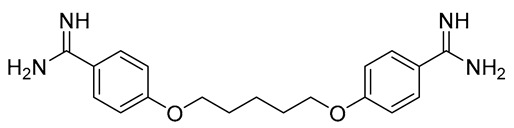 **Pentamidine**	Parenteral (I.M.),	Interferes with DNA synthesis and modifies the morphology of kinetoplast	Pain, headaches, nausea, vomiting, myalgia, temporary hyperglycemia, and dizziness	Hypertension, electrocardiographic alterations, tachycardia, and hypotension	[11]
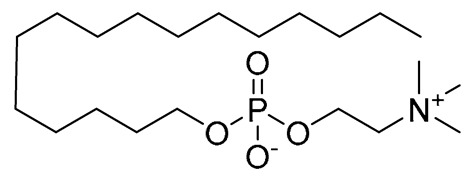 **Miltefosine**	Oral for VL	Associated with leishmanial alkyl-lipid metabolism and phospholipid biosynthesis	Nausea, vomiting, and diarrhea	Elevated creatinine, the toxicity of the kidneys and liver, and teratogenicity	[12]
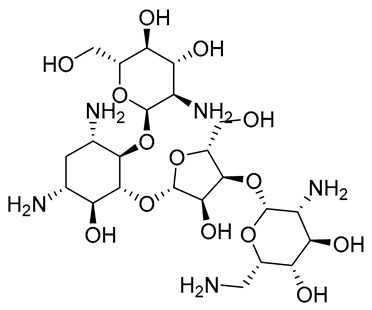 **Paromomycin**	Topical for CL Parenteral for VL	Inhibits the protein biosynthesis in sensitive *Leishmania* parasite	Erythema, pain, and allergic contact dermatitis [13]	Liver toxicity, internal ear damage, erythema, discomfort, edema, and renal toxicity	[13]

## 2. Methods

### 2.1. Data Collection

Data were collected from the previously reported peer-reviewed research papers, and case reports describe the CL treatment based on non-drug options. Keywords, including combinations of “cutaneous leishmaniasis” and each of the treatment options shown in Section 3 of this review, were used for locating the target study in various international databases, including PubMed, SciFinder, and Google Scholar.

### 2.2. Limitations

Treatment approaches associated with significant skin damage are infrequently documented in the scientific database, and those that have no significant treatment effect on cutaneous leishmaniasis were not covered in this review. Alternative treatments that leave unsightly scarring cannot be proposed, and informed consent should be adequately prepared. Additionally, based on the goal of this review, the data were designed, in most therapeutic strategies, in tabular form to enable comparison between protocols followed and achieved outcomes, which we think will aid in the process of finding the best species-specific medications with minimal side effects, low costs, and greater efficacy.

### 2.3. Statement of Ethics Compliance

This article is based on previously published work and does not contain novel data or information related to human or animal studies.

## 3. Cutaneous Leishmaniasis Non-Drug Treatment Options

### 3.1. Cryotherapy

In cryotherapy, the lesions are heated to extremely low temperatures, such as −196 °C using liquid nitrogen (N_2_) or—80 using liquid carbon dioxide (CO_2_) [14]. Cryotherapy destroys affected tissues and kills amastigotes by causing intracellular and extracellular ice crystal production and alterations in the cell membrane [15]. Cryonecrosis results in the release of antigenic compounds that stimulate immune responses and repair further lesions [16]. Although painful, it is a simple, inexpensive, and rapid procedure devoid of the need for local anesthesia [16]. With few reported cases of burning and the possibility of secondary infections after treatment with cryotherapy [17], it can be a helpful alternative to chemotherapy for the treatment of CL. According to studies, cryotherapy provides satisfactory treatment results and lowers relapse rates in most cases, with varied degrees of success depending on the causative parasite [17]. CL caused by *L.* (*L*) *major* exhibited ~84% healing rate after 1–4 liquid N_2_ treatment sessions, while the remaining lesions were treated with additional 1–3 sessions, leaving barely noticeable scarring and no relapses. Additionally, *L.* (*L*) *infantum*, *L.* (*L*) *tropica*, and *L.* (*L*) *aethiopica* [18] exhibited excellent thermosensitivity, whereas *L.* (*Viannia*) *braziliensis* exhibited low thermosensitivity [19]. Cryotherapy was beneficial in treating *L.* (*L*) *donovani*-related CL in Sri Lanka, but the condition was followed by scarring, depigmentation, and ulceration. In Ethiopia, *L.* (*L*) *aethiopica* was treated with effectiveness comparable to that of sodium stibogluconate. [20,21]. The physical location and size of the lesion, in addition to the causative parasite, have been found to have a significant impact on the treatment response; those with smaller lesions (1 cm) exhibit improved outcomes [22]. It is important to note that a meta-analysis based on clinical studies found that cryotherapy for the treatment of the genus *Leishmania* is as successful overall as antimonials [23]. Although there has been relatively little prior research on New World CL, in general, cryotherapy appears to be effective against a variety of CL infections caused mainly by Old World CL species [22].

Additionally, improved results have been seen when chemotherapeutic drugs are administered intralesionally along with cryotherapy. The combination of cryotherapy with intralesional sodium stibogluconate (ISSG) was found to be very efficacious, resulting in 100% healing of CL lesions, whereas the combination with itraconazole resulted in about 80% improvement in CL lesions with low hepatotoxicity risk as a result of a reduced dose of itraconazole [24,25].

### 3.2. Photodynamic Therapy

Today, photodynamic therapy (PDT) is one of the novel developments to treat CL [26]. The term photodynamic therapy refers to the use of photo-excitable dyes in combination with particular wavelength frequencies to cause the generation of reactive oxygen species (ROS), which in turn drive the parasites to be photodynamically inactivated [27]. The photosensitizer accumulates specifically in the target tissue, and when exposed to visible light of the right wavelength, it activates molecular oxygen to produce reactive oxygen species [28]. Leishmania species have been the primary targets for PDT. To enable the complete deactivation of *Leishmania*, the dyes uroporphyrin and phthalocyanines are used [29]. Other photosensitizer dyes, including methylene blue, chloroaliminium methyl aminolevulinate (MAL), aminolevulinic acid, hematoporphyrin, curcumin, and hypericin, have been found to be effective on the amastigote and promastigote forms of *Leishmania* species in several studies (Table 2) [30].

The primary benefit of this treatment is that the dye can preferentially concentrate inside the parasite before the ROS-inducing wavelengths are applied. This enables the pathogens to be effectively destroyed without endangering the host tissue. Reported adverse effects include mild pruritus, burning, erythema, oedema, local inflammation, residual hyperpigmentation, and pain [31,32].

**Table 2 tropicalmed-08-00275-t002:** CL photodynamic treatments and how long it takes to get the optimal results.

The Parasite	The Treatment Protocol	Weeks to Achieve the Best Outcome	Ref.
*L.* (*L*) *major*	PDT with topical 5-aminolaevulinic acid (5-ALA) and red light at 633 nm (100 joule/cm^2^ (J/cm^2^), 4 sessions).	A total of 4 weeks until achieving a complete response in 93.5% of lesions.	[31]
*L.* (*L*) *major*	PDT with topical 5-ALA and red light, (570–670 nm), (100 J/cm^2^, 4 sessions).	Four weeks until achieving a complete cure.	[32]
*L.* (*L*) *major* and *L.* (*L*) *tropica*	Solar photoprotector (SFP) and daylight-activated PDT with topical MAL. Exposure to daylight for 2.5 h (dose was not determined, <8 sessions).	Complete cure in 89%.	[33]
*L.* (*L*) *major*	PDT with 5-ALA and red light, (37 J/cm^2^, 24 sessions).	Twelve weeks until achieving complete healing.	[34]
*L.* (*L*) *tropica*	PDT with 5-ALA and red light, 633 nm (75 J/cm^2^, 3 sessions).	Eighteen weeks until achieving a complete cure. Complete resolution after the third session.	[35]
*L.* (*L*) *tropica*	PDT with topical MAL and red light, 635 nm (100 J/cm^2^, 3 sessions).	Six weeks until achieving a complete response.	[36]
*L.* (*L*) *tropica*	PDT with 5-ALA and red light, 630 nm (37 J/cm^2^, 5 sessions).	Five weeks until achieving a complete recovery.	[37]
*L.* (*L.*) *tropica*	PDT with MAL and red light, 630 nm (dose was not determined, 8 sessions).	Seven weeks until achieving a complete cure.	[38]

### 3.3. Thermotherapy

*Leishmania* species that cause cutaneous illness cannot survive, grow, or multiply within the host’s macrophages in environments hotter than 39 °C [39]. Both hot water baths, laser therapy, ultrasound, infrared light and microwaves, radiofrequency [8], and exothermic crystallization thermotherapy are used to generate a non-localized heat that may destroy the surrounding tissue around the targeted cutaneous lesion [8,40]. Thermotherapy was found to be safe and effective and provided better treatment outcomes for CL than chemotherapeutic drugs, according to previous research, which is covered in more detail in the following.

#### 3.3.1. Hot Water Baths

In hot water baths, CL lesions receive heat through hot water cushions that surround them, maintaining a temperature of 39–41 °C for at least 20 h over several days of treatment [41]. This method was applied to three diffuse CL patients, and every single patient experienced a complete recovery after a minimum cumulative treatment time of 20 h. Given that diffuse cutaneous leishmaniasis responds poorly to conventional drug treatment, this outcome is encouraging. However, further research should be done on this technique because it was successful in treating only the diffuse CL form of leishmaniasis [41].

#### 3.3.2. Laser Therapy

Treatment of CL remains difficult for doctors despite extensive research and the development of several medicines for the disease that include both local and systemic approaches. For many years, dermatological disorders such as rosacea, vitiligo, and acne have been treated with laser therapy. Today, one of the laser’s suggested applications is the therapy of CL. The effectiveness of lasers is compiled in (Table 3) [42,43,44,45,46,47,48,49,50,51,52,53,54,55,56,57,58,59] and includes examples of continuous and fractional CO_2_, argon, pulsed dye laser (PDL), erbium glass, and neodymium-doped yttrium aluminum garnet (Nd: YAG). According to research, several mechanisms explain why laser therapy is beneficial for curing CL lesions. Utilizing a CO_2_ laser, selective photothermolysis, which is 2–4 times more common as compared with cryotherapy or electrosurgery, allows a laser beam to virtually exactly target and remove necrotic tissue [47,48,49,50,51,52,54,55,60,61,62]. The additional passive pathways include cytokine modification, immune system stimulation, and induction of an inflammatory response [47,57]. At this point, we list the various laser types and their range of wavelengths as important tools for treating CL. Physicians may be able to select the most effective laser treatment for CL using the information acquired about the various laser systems in Table 3.

#### 3.3.3. Ultrasound Therapy

CL was treated with ultrasound in 1987; Leishmaniasis was treated with ultrasound using a machine with a 2 cm handheld applicator; the ultrasound intensity was 1.5–3 watts (W)/cm^2^, and ThermoProbe was adjusted at a target temperature of 42 °C. In total, 22 out of 28 lesions (78.5%) completely disappeared within 5–10 weeks, with 2–3 ultrasound sessions per week. Ultrasound was well accepted by the patients, and no negative effects were noted, which encouraged the authors to conclude that CL is safely treated with heat therapy utilizing ultrasound [63]. However, given the shortage of knowledge regarding the use of ultrasound for the treatment of CL, more research is recommended.

#### 3.3.4. Infrared and Microwave

According to some theories, infrared therapy works by either inhibiting the parasite or stimulating an immune response [64]. Infrared radiation (5 min) at 55 °C was administered to 178 individuals with CL. The bulk (162) of the patients were treated in a single session, followed by 15 patients who required two sessions, and the remaining patients required three sessions. BALB/C mice (20) were used to study the effects of microwave versus infrared radiation and their mixture on L. (*Leishmania*) *major* skin lesions. A 150 W infrared device with λ = 890 nm and a 600 W microwave device with a 2.45 gigahertz (GHz) frequency. Infrared radiation proved more effective than microwaves at preventing the growth of ulcers [65].

Infrared thermotherapy was used three times a week to treat 35 CL patients (53 lesions) at the Imam Reza Hospital in Mashhad, Iran. The lesions were warmed to 45 degrees centigrade for two rounds of 10 min. Complete (90–100%), good (50–89%), and poor responses (less than 50% size decrease) were the three categories used to describe the treatment outcomes. After three months, 13 (24.5%) lesions were completely healed, and 31 (58.5%) and 9 (17%) other lesions responded well or poorly [66]. An 890 nm infrared laser was found to boost the production of nitrous oxide and hasten the healing of skin lesions [66].

In a case-controlled clinical trial, 35 patients with one or more CL lesions received microwave radiation treatment every two weeks for a total of eight weeks. After 1–4 sessions of microwave therapy, 85.33% of lesions were clinically healed; however, only 20.83% of lesions in the control group showed some healing. Microwave heat therapy is a novel, extremely effective treatment for CL with no reported adverse effects [67].

#### 3.3.5. Handheld Exothermic Crystallization Thermotherapy

A gadget composed of a flexible metal disc and supersaturated sodium acetate solution enclosed in a sealed plastic pouch. An exothermic liquid-to-solid phase change reaction, with a maximal temperature of 52 °C ± 2 °C, will occur with disc flexing and release heat [68]. CL was first treated with this technique in Peru. Seven days of treatment and six months of follow-up were given to 25 CL patients. With only two cases of second-degree burns, 68.4% of patients demonstrated complete recovery [69]. The second investigation was carried out in Pakistan. With a starting temperature of 51.6 °C for 3 min, the patients received treatment for 7 days. Out of the 23 patients who finished the research and were followed up for 6 months, 19 (83%) had fully recovered [68]. Recently, a randomized controlled clinical trial was conducted in Sri Lanka from January 2017 to January 2018 on 40 CL (mostly caused by *L.* (*L*) *donovani*) treatment failures with IL-SSG. The trial involved two arms: radio frequency-induced heat therapy (RFHT) using a ThermoMedTM device (Model 1.8, Thermosurgery Technologies, Inc., Phoenix, AZ) and thermotherapy using a handheld exothermic crystallization thermotherapy (HECT-CL). At days 90 (initial cure rate) and 180 (final cure rate) after treatment, intention-to-treat cure rates were computed. Group receiving radio frequency-induced heat therapy: 100% (20/20) of patients were initially cured, and 95% (19/20) were eventually cured, with one patient relapsing. The initial and ultimate cure rates (16/20) for the HECT-CL group were both 80%, with no relapses and one participant being dropped from the trial [70]. In contrast, in Pakistan, 56 CL patients caused by *L.* (*L*) *tropica* received handheld exothermic crystallization thermotherapy heat pack therapy for 3 min over 7 continuous days and experienced a 91% failure rate [71]. The authors suggested this high rate of treatment failure was due to *L.* (*L*) *tropica*’s low heat sensitivity and slower spontaneous healing. Although handheld exothermic crystallization thermotherapy is a safe, low-cost CL treatment option, these unexpected treatment results point to species-specificity that may call for additional research in larger cohorts with different parasites to highlight the sensitive species.

#### 3.3.6. Radiofrequency

Radiofrequency-based thermotherapies were developed as a way to target leishmanial lesions more precisely without damaging surrounding tissue, resulting in a higher-quality treatment with fewer side effects [28]. Examples of CL treatment with radiofrequency-induced heat therapy are listed in (Table 4).

To generate heat using a current field radiofrequency generator, this treatment entails the controlled and localized delivery of radiofrequencies into lesions for 30–60 s while under local anesthetic. The heat produced permeates the upper dermis, resulting in a secondary burn that kills sick tissue while doing the least amount of harm to good underlying tissue. According to some short-term (4–5 months) follow-up trials, RF heat (RFH) therapy is just as effective as antimonials in treating CL [76]. The reasons underlying RFH therapy’s great success in the treatment of CL are unclear, despite the evidence to the contrary.

Direct heat killing of parasites is one such way. Alternately, tumor necrosis factor (TNF)-activity induction in response to heat and/or by enhancing the current immune response due to parasite antigen release from damaged tissue could quickly upregulate a protective Th1 response [83]. There have been at least three tools used for radiofrequency heat treatment, including the RDM handheld radiofrequency heat generator, the ThermoMed device, and the Ellman radiofrequency heat generator (Table 4).

### 3.4. Traditional Non-Drug Therapy

Leech and cauterization therapy were the most common non-drug treatments in traditional medicine. Leech therapy, also known as hirudotherapy, is a type of unconventional medical approach that uses leeches (Figure 1) that feed on human blood. To benefit from the leech, one or more leeches are connected to the skin of the troublesome location. Leeches pull blood from wounded tissues to expel blood and morbid humor. Fresh blood then perfuses the tissues, hastening the healing process. More than 20 known bioactive compounds, including antistasin, eglins, guamerin, hirudin, saratin, bdellins, complement, and carboxypeptidase inhibitors, have been found in the saliva of leeches. Leeches’ saliva acts as an anticoagulant and has thrombolytic, anti-inflammatory, platelet inhibitory, anticoagulant, and thrombin-regulating actions in addition to degrading extracellular matrix and having antibacterial effects. On the other hand, it has been demonstrated that this chemical contains anesthetic and antibiotic-like characteristics. Additionally, elements of leech saliva can function as vasodilators and improve blood flow and circulation in the targeted area [84]. A case study reported the CL treatment of two cases, a 56-year-old man and a 43-year-old woman, by leech therapy. The male patient underwent leech therapy four times at weekly intervals. After two months, the lesion was fully cured. The healing process was observed five months after the original leech therapy session, and the lesions showed no evidence of relapsing. The woman underwent five leech treatments, separated by two-week and one-month intervals. After six months, the lesion was completely gone. Her condition was monitored for a further 1.5 years, and there had been no sign of a recurrence [85].

To hasten the healing process, some doctors suggested kay (cauterization) of the lesions and tissue removal. The following step was advised: cautery twice, depending on the depth of the lesion, after rubbing the burned area with a rough fabric to remove the burned tissues. The doctors were cautioned to be careful to protect the nerves, tendons, and ligaments because this treatment may potentially go down to the bone and be quite painful. Other suggested methods for treating CL include transferring the patient to cooler locations and using an enema to cleanse the body of internal wastes [86]. A three-arm, phase IIb, randomized, and controlled clinical trial was conducted on CL 87 patients who were infected with *L.* (*L*) *tropica* or *L.* (*L*) *major*. Intradermal sodium stibogluconate was administered to Group I’s 24 participants. Group II’s 32 participants received high-frequency electro-cauterization (HF-EC) and then MWT with 0.045% pharmaceutical chlorite (*Sodium chlorosum,* DAC N-055), while Group III’s 31 participants received moist-wound-treatment (MWT) with polyacrylate hydrogel with 0.045% DAC N-055 in basic crème alone. The electro-cauterization method was found to be more efficient for CL treatment, where complete epithelialization before day 75 was found only in 15 (of 23; 65%) patients in Group I, 23 (of 23; 100%) patients in Group II, and 20 (of 23; 87%) patients in Group III, according to the per-protocol study of 69 (79%) patients [87]. In another study, the EC plus MWT therapy hastened the healing of wounds as compared with the DAC N-055 [88].

## 4. Conclusions

Depending on the *Leishmania* species, significant variations in CL treatment efficacy are observed, with a potential connection to the endemic area. Cryotherapy is a quick, easy, and economical method that does not require local anesthesia, even though there have been few reports of burning and there is a chance of secondary infections. All *Leishmania* parasites, except for *L.* (*V*) *braziliensis*, demonstrated excellent thermosensitivity. Results have improved when chemotherapeutic medications are delivered intralesionally in conjunction with cryotherapy.

Thermotherapy has fewer side effects than other CL therapies and should be encouraged whenever feasible. More significantly, the heat impact effectively inhibited several *Leishmania* species and was useful for CL cases whose chemotherapy treatments had failed. Combinations were successful in improving therapy outcomes; the combination of antimonial drugs with cryotherapy is more successful than antimonials alone because the cumulative effect may increase the antimonials’ effectiveness.

When photo-excitable dyes are combined with specific wavelength frequencies during photodynamic treatment, reactive oxygen species (ROS) are formed, which promotes high rates of photodynamic eradication of the infections, except for a few unfavorable effects such as moderate pruritus, burning, erythema, persistent hyperpigmentation, and discomfort.

In summary, due to the different systemic toxicities, elevated cost, and drug resistance associated with long-term use of conventional drug treatment of CL, the compiled examples of the different CL alternative therapies in this review may enable doctors to choose the most efficient treatment strategy with the least adverse effects, utilizing the knowledge they have gained about the various options.

## Figures and Tables

**Figure 1 tropicalmed-08-00275-f001:**
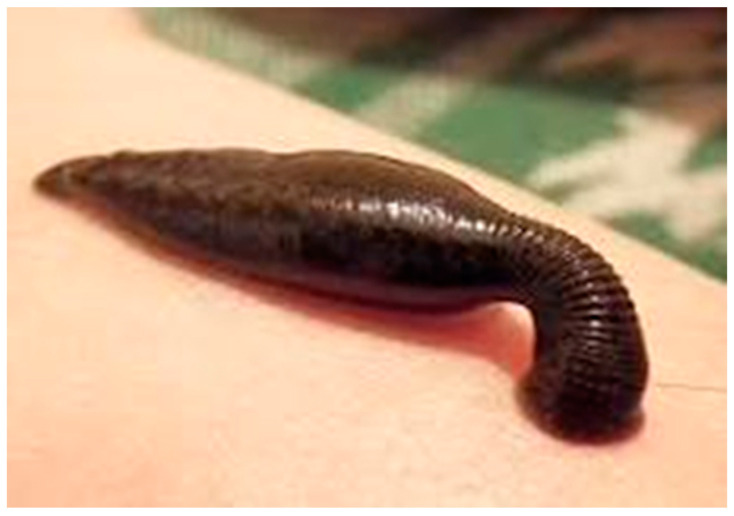
Image of Leech sucking a blood meal.

**Table 3 tropicalmed-08-00275-t003:** List of reported CL laser treatments and achieved outcomes.

Laser Type	Cases	Treatment Session	Achieved Outcomes	Ref.
Fractional CO_2_ laser + paromomycin.	A 16-year-old female with several non-tender, non-healing CL wounds on her bilateral upper and lower extremities.	Two sessions at one-month interval.	- Multiple small lesions were treated successfully with oral fluconazole and topical paromomycin. - The largest lesion (measuring 2 × 0.8 cm in size) was successfully treated.	[42]
Fractional CO_2_ laser, assisted by topical rifamycin.	One child.	Two sessions at one-month intervals of fractional CO_2_ laser, followed by topical application of rifamycin for three days.	- The lesions were resolved with minimal scarring.	[43]
Fractional ablative CO_2_ laser followed by ISSG application.	Ten children.	Patients were treated with fractional ablative carbon dioxide laser followed by immediate ISSG.	- A total of 90% of patients showed satisfactory to exceptional clinical cure with high patient satisfaction and tolerance and absence of adverse effects after and during treatment; - One girl diagnosed as positive *Leishmania* (*L*) *tropica* failed to respond.	[44]
Fractional Ablative CO_2_ laser followed by ISSG.	A total of 20 patients/181 lesions.	- Control group: ISSG; - Study group: Fractional CO2 laser treatment followed by ISSG.	Compared with ISSG, fractional CO_2_ laser treatment followed by ISSG application is less painful and produces a superior final cosmetic result.	[45]
Comparison between the effects of: - Glucantime alone; - 50% topical trichloroacetic acid (TCA) + Glucantime; - Fractional CO_2_ laser + glucantime.	Ninety patients in three groups: - Glucantime alone: 30; - TCA + fractional CO_2_ laser: 30; - Intralesional glucantime + fractional CO_2_ laser: 30.	- The duration of treatment was: glucantime 6.8 ± 1.7 weeks; - Topical TCA plus glucantime 5.2 ± 1.0; - Fractional laser plus glucantime groups: 6.3 ± 3.0 weeks; - The laser treatment was 2 sessions at 2 weeks intervals, and the intralesional injection of glucantime was twice weekly for 8 weeks.	Complete healing in: - A total of 38.5% of patients were treated with glucantime alone; - A total of 90% with glucantime + TCA; - A total of 87% with glucantime + fractional CO_2_ laser.	[46]
Pulsed dye laser (PDL).	A total of 17 patients with 81 lesions: - PDL group: 49 lesions; - Intralesional meglumine antimoniate (MA) group: 39 lesions.	- A total of 42 lesions were treated with one or two passes of PDL fortnightly; - A total of 39 lesions in the same patients were treated with intralesional MA weekly.	- Almost 66.7% of laser-treated lesions were cured after the first therapy, and more than 90% were cured after the second. One patient was healed after the third treatment, and the final one after the fourth. - All MA-treated lesions were cured after 3–8 sessions.	[47]
	A total of 25 lesions in 12 patients.	- All lesions were treated with a single pass of PDL.	- A total of 13 lesions showed rapid response after 3 sessions. - The remaining 12 lesions required 4 sessions.	[48]
	Three patients: - A 14-year-old girl with an ulcerative and erythematous nodular lesion on her left cheek; - A 43-year-old woman presented an erythematous popular lesion on her right cheek; - A 60-year-old woman presented an erythematous popular lesion on her cheek.	- PDL 595 nm was used at 3 ms duration, 7 mm spot size, and energy: 8 j/cm^2^; - PDL 595 nm was used at 3 ms duration, 10 mm spot size, and energy: 8 j/cm^2^; - PDL 595 nm was used at 3 ms duration, 7 mm spot size, and energy: 8 j/cm^2^.	- After the first session, the erythematous and telangiectasic lesions began to improve, and by the third, they were entirely gone.	[49]
Continuous CO_2_ laser.	- Study group: 183 lesions in a total of 123 individuals (68 female and 55 male); - Control group: 110 patients (with 250 lesions).	- The study group was treated with the CO_2_ laser (100 W and the pulse width was 0.5–5 s); - The control group was treated with glucantime 50 mg/kg/day for 15 days.	- A single session of the CO_2_ laser was 1.12 times more successful than glucantime. It also had fewer side effects (4.5% vs. 24%) and caused healing to take less time (1 month vs. 3 months).	[50]
	- A total of 108 patients: 101 of the 108 patients were young or middle-aged, and 89 (82%) of them were males.	Patients were administered a local anesthetic and treated by a focused laser beam (surface power density was about 2.3–3.0 kW/cm^2^. Finally, the defocused laser beam is directed at the wound (laser surface power density: 200–400 W/cm^2^) until the entire wound surface becomes covered with a thin light-brown film.	A concentrated laser beam could vaporize individual leishmaniasis lesions. In cases of numerous lesions, the largest and most infected ulcers were removed in a single session (up to 5–6 foci), and the remaining lesions were removed as soon as the first wounds started to epithelialize.	[51]
	Twenty-four patients with lupoid cutaneous leishmaniasis for more than one year.	- CO_2_ laser (maximum power was 100 W and the pulse width was 0.5–5 s);	- A total of 21 patients (13 women and 8 men) were then monitored for a full year. Only 2 of these patients (9.5%) experienced therapy failure, and 19 (90.47%) were disease-free.	[52]
	- Laser group: 80 patients with 96 lesions; - Combined therapy group: 80 patients with 95 lesions.	CO_2_ laser (maximum power of 30 W and pulse duration of 0.01–1 s, and mode of a continuous wave); Cryotherapy (twice weekly) with intralesional meglumine antimoniate weekly, until complete cure or up to 12 weeks.	- Laser group: 93.7% (89/96); - Combined therapy group: 78% (74/95).	[53]
Erbium glass laser.	A total of 14 patients/20 lesions.	Weekly sessions with the Palomar 1540 nm erbium glass fractional laser using a handpiece with a 10 mm spot size, four passes of 50–70 mJ/cm^2^ fluence, and a pulse length of 10 ms.	- A total of 6 patients (50%) improved at 6 weeks; - A total of 11 patients (91.7%) healed at 12 weeks; - No recurrences up to 6–12-month follow-ups	[54]
Neodymium-doped yttrium aluminium garnet (ND/YAG) laser.	A total of 16 patients were treated simultaneously, where one lesion was treated with glucantime and another with ND/YAG laser.	Average glucantime sessions = 7.31 ± 4; Average laser therapy sessions = 2.56 ± 0. 9.	ND/YAG laser therapy for CL led to complete recovery of patients in a shorter time with fewer complications than glucantime medication.	[55]
Low-level laser therapy.	Thirteen patients.	A Diode laser probe with λ 820 nm was used, followed by a cluster probe. Three sessions weekly for a total of ten sessions, the dose was: I. Diode laser probe with an energy density of 48 J/cm^2^ 30 s; II. Cluster probe with an energy density of 9.6 J/cm^2^ for 2 min.	A total of 92.3% of the patients who received treatment had excellent outcomes. The difficulties were minor and temporary.	[56]
	A total of 53 patients/123 lesions. Divided into two groups: - Group 1: patients with early-stage CL (papule of size ≤ 1 cm); - Group 2: patients with high-grade CL (vesicle formation, ulceration, and superadded infection of size > 4 cm).	Four sessions at one-week intervals.	- A 10-month follow-up revealed complete response in 91% and partial response in 9% of patients with early-stage CL; - No response was observed in patients with high-grade disease.	[57]
Intravenous laser blood irradiation (ILBI).	A total of 40 patients with one or more wounds of at least 5 cm diameter and a swollen and purulent mass around the wound. - Fifteen patients underwent traditional therapy; - Twenty-five patients underwent traditional therapy in combination with ILBI therapy.	Ten days of ILBI therapy using the Matrix-VLOK device, continuous radiation mode for 15 min, alternating VLOK radiating heads at a wavelength of 0.63 microns and radiation energies of 1.5–2.0 mV.	- CL treatment with ILBI showed quicker mortality of leishmaniasis with well-tolerated and faster wound healing, and a quicker progression to the stage of scarring.	[58]
	A total of 40 patients, each having one or more wounds that are at least 5 cm in diameter.	A total of 25 patients received intravenous laser therapy in addition to regular medical treatment for 10 days, while 15 patients received regular medical treatment alone.	It has been demonstrated that ILBI therapy is successful as a painless treatment for patients undergoing clinical and pathogenetic therapy.	[59]

**Table 4 tropicalmed-08-00275-t004:** List of radiofrequency-based heat therapy treatments for CL and achieved outcomes.

Parasite	Patients/Groups	The RF Procedure	Results/Healing Rate	Ref.
*L.* (*V*) *braziliensis* *L.* (*L*) *mexicana*	- Systemic glucantime: (22 patients); - RF group: (22 patients); - Placebo group: (22 patients).	RF generated heat (50 °C for 30 s, 3 times at 7-day intervals).	- Systemic glucantime: 73%; - RF: 73%; - Placebo: 27%.	[72]
*L.* (*L*) *tropica*	- One patient (ten lesions).	RF generated heat (50℃ surface temperature for 30 s, using a handheld RF heat generator).	- Six weeks after the initial course of therapy, all lesions were entirely granulated, and six months later, every lesion was fully healed.	[71]
*L.* (*L*) *mexicana*	- RF group: 201 patients.	Localized current field (LCF-RF) generated heat (50 °C for 30 s, one time).	A total of 90% healing rate.	[73]
*L.* (*L*) *tropica*	- IL-SSG group: 148 patients; - Intramuscular sodium stibogluconate (IM-SSG) group: 144 patients; - RF group: 139 patients.	RF generated (50 °C for 30 s), using ThermoMed 1.8 RF generator).	- IL-SSG group: 70/93 (75.3%); - IM-SSG group: 26/58 (44.8%); - RF group: 75/108 (69.4%).	[74]
Anthroponotic	- RF group: 57 patients with 83 lesions; - IL glucantime group: 60 patients with 94 lesions.	RF generated heat (50° for 30 s, 4 times at 7-day interval).	- RF group: 80.7%; - IL glucantime group: 55.3%	[75]
Not identified	- RF group: 47 patients; - MA group: 59 patients.	LCF-RF generated heat (50 °C for 30 s, one time), using a ThermoMed 1.8 LCF-RF generator.	- RF group: 100%; - MA group: 19%	[76]
*L.* (*L*) *major*	- RF group: 27 patients; - Systemic SSG group: 27 patients.	RF generated heat (50 °C for 30 s, one time), using ThermoMed 1.8 generator.	- RF group: 73%; - SSG group: 59%.	[77]
*L.* (*L*) *tropica*	- RF group: 195 patients; - IL glucantime group: 195 patients.	RF generated heat (50 °C for 30 s, one time), ThermoMed 1.8 generator.	- RF group: 82.5%; - IL glucantime group: 74%.	[76]
*L. panamensis L. (V) braziliensis* *L.* (*L*) *amazonensis L.* (*L*) *mexicana* *L.* (*L*) *infantum*	- RF group: 149 patients; - Systemic MA group: 143 patients.	RF generated heat (50 °C for 30 s, one time), ThermoMed 1.8 generator.	- RF group: 64%; - MA group: 85%.	[78]
*L.* (*L*) *tropica*	- RF group: 50 patients; - IL-SSG group: 50 patients (7 sessions).	LCF-RF generated heat (50° for 30–60 s), using RF ThermoMed 1.8 generator.	- RF group: 98%; - IL-SSG group: 94%.	[79]
*L.* (*V*) *panamensis* *L.* (*V*) *braziliensis*	- RF group: 149 patients; - Miltefosine group: 145 patients.	RF generated heat (50 °C for 30 s), using ThermoMed 1.8 generator.	- RF group: 59%; - Miltefosine group: 59%.	[80]
*L.* (*L*) *donovani*	- RF group: 93 patients; - IL-SSG group: 115 patients.	ThermoMed Model 1·8.	- RF group: 65.9%; - IL-SSG group: 59.4%.	[81]
Not identified	- RF group: 15 patients.	RF generated heat (50 °C for 30 s, one time), ThermoMed 1.8 generator.	- RF group: 85.7%.	[82]

## Data Availability

Questions from the survey are available with the first author (M.A.A.O.).

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
