# Peer review of "Alternative Non-Drug Treatment Options of the Most Neglected Parasitic Disease Cutaneous Leishmaniasis: A Narrative Review"

_tropicalmed, 2023, doi:10.3390/tropicalmed8050275_

Round 1

Reviewer 1 Report

In the manuscript submitted to Molecules (id nº tropicalmed-2378823), entitled “Alternative Non-Drug Treatment Options of the Most Neglected Parasitic Disease Cutaneous Leishmaniasis: A Narrative Review”, by Orabi et al. the authors propose to compile information on non-conventional (mainly non-drug-based) treatments for cutaneous leishmaniasis. Such a compilation is interesting and useful for those working both within and outside the field. First of all, it is noticeable that the article was written by non-English native speakers. In several instances the message is not clear; therefore, I suggest you to ask someone proficient in English to read the manuscript and correct the English. Additionally, I think this Review would benefit from a subjective perspective. I find it very descriptive without any critical input from the authors. Please check below some points that must be addressed before I consider recommend this paper for acceptance.

Major points:

1-     As per the convention, Latin species names must be italicized. However, throughout the text, Leishmania… appears non-italicized. Please revise for accuracy. Also on this subject, only the species name is  capitalized, as per the convention. Therefore Mexicana should read mexicana.

2-     Plant extracts are used for millennia to treat diseases. Therefore, modern is not a very good adjective to use in this context.

3-     The Introduction should be “general” to contextualize the review. Therefore, I don’t think you should mention specific studies like you do from line 57 to 76.

4-     I would not use the heading results. It is not appropriate in the context of a Review Article, at least in my opinion.

5-     The authors comment a lot on the effectiveness. However, in several instances it is not clear the comparisons done. Compared with drug treatment? How long would the lesions would take to heal in the absence of treatment? Please add these details to throughout the paper. E.g:

a.      Hot baths;

b.     Ultrasound therapy;

c.      Traditional therapies;

d.    

6-     In the context of cristalization therapy you mention contradictory results. Therefore, you should discuss more on this. Is it effective or not? Is it species specific?...

Minor points:

1-     The use of English language should be polished. Particularly, there are many examples of subject verb disagreement throughout the text. Here are just a few examples – there are a lot more throughout the text:

a.      E.g. Page 1, line 22: “regions of about 90 different countries…” does not read well.

b.     E.g. Page 4, line 130: ROS is plural, so the verb should read “drive” and not “drives”.

c.      E.g. Page 4, line 145: should read “optimal”.

2-     Line 96: You mention extremely low temperatures and then exemplify with 196 C. I think this is a typo since you are highlighting a temperature well above the water boiling temperature. Please correct for accuracy.

3-     Line 137: phthalocyanines and uroporphyrin should not be mentioned as other since they were highlighted before.

4-     As per the conventions abbreviations must be defined at first appearance.

5-     Line 191: what do you mean with distracting?

6-     Lines 234-236: a reference is missing.

7-     Line 248: should read traditional non-drug therapy.

Some sentences are hard to understand. It is clear the paper was written by non-native speakers. I would recommend an extensive revision of the paper for the sake of accuracy and readability.

Author Response

The responses to the reviewer's comments are shown in the attached file.

Reviewer 2 Report

The current study performed a narrative review on the alternative non-drug treatment options of cutaneous leishmaniasis (CL), one of the NTDs, caused by Leishmania parasites. As the authors summarized that the compiled examples of the different CL therapies in this review may enable physicians to choose the most efficient treatment strategy with the least adverse effects utilizing the knowledge they have gained about the various options, this paper provides useful information for the management of CL cases in the Old and New World endemic areas of the disease. The study is well organized and designed as a narrative review; besides the following points are better to be further considered:

Major comments:

Regarding the scientific names of the causative agents of CL, Leishmania parasites, I recommend the use of subgenus names, Leishmania (Leishmania) and Leishmania (Viannia); e.g., L. (L.) major, L. (L.) aethiopica, and L. (V.) braziliensis, L.(V.) panamensis, and etc. in the text and Tables including abstract, since it has been frequently reported that  the subgenus difference affects the treatment results and relapse rates in CL cases with varied degrees of success according to the causative parasite.

 Minor comments:

- Throughout the text and Tables, replace new world-CL by New World-CL, and old world by Old World-CL.

-Line 86, replace “coetaneous” by “cutaneous”.

- Line 127, replace “to destroy” by “to treat”.

- in Table 4, replace L. Mexicana by L. (L.) mexicana, and provide subgenus (L.) or (V.) for each parasite.

- Arrange adequately Ref. nos. 58 and 84 for English versions, and thoroughly check all the references listed, following the journal’s instruction.

Major comments:

Regarding the scientific names of the causative agents of CL, Leishmania parasites, I recommend the use of subgenus names, Leishmania (Leishmania) and Leishmania (Viannia); e.g., L. (L.) major, L. (L.) aethiopica, and L. (V.) braziliensis, L.(V.) panamensis, and etc. in the text and Tables including abstract, since it has been frequently reported that  the subgenus difference affects the treatment results and relapse rates in CL cases with varied degrees of success according to the causative parasite.

Minor comments:

- Throughout the text and Tables, replace new world-CL by New World-CL, and old world by Old World-CL.

-Line 86, replace “coetaneous” by “cutaneous”.

- Line 127, replace “to destroy” by “to treat”.

- in Table 4, replace L. Mexicana by L. (L.) mexicana, and provide subgenus (L.) or (V.) for each parasite.

- Arrange adequately Ref. nos. 58 and 84 for English versions, and thoroughly check all the references listed, following the journal’s instruction.

Author Response

The responses to the reviewer comments is shown in the attached file

Reviewer 3 Report

- An historical article about Leishmania worldwide should be added in the intro section: Nazzaro G, et al. Leishmaniasis: a disease with many names. JAMA Dermatol. 2014; 150(11):1204. doi: 10.1001/jamadermatol.2014.1015. PMID: 25389793.

- Reference 6 about drug resistence dates back to 2006. Do you have a more recent one? Which drug have the highest risk of resistence?

- In Table 1 for topical paromomycin "allergic contact dermatitis" is missing among the mild to moderate effects (Veraldi S, et al. Allergic contact dermatitis caused by paromomycin. Contact Dermatitis. 2019; 81(5):393-394)

- Methods should be better described. Which keywords did you search? Which kind of papers?

- page 3. paragraph 2.2. please correct cutaneous

- In general, in this narrative review it is not clear if the proposed treatments are supported by a consistent bibliography. For example, in Table 2 the reported experiences are single case reports or retrospective case series

- In a limitation paragraph, in the discussion, it should be stated that "alternative" treatments leaving unsightly scarring cannot be proposed or an informed consent should be adequately prepared.

- In conclusion, why a doctor should take into account to use an "alternative" treatment rather than a pharmacological one?

Author Response

The responses to the reviewer comments are shown in the attached file

Round 2

Reviewer 1 Report

My concerns were addressed.

Some improvements can still be done. I assume the journal typewriters will take care of that.

Reviewer 3 Report

The manuscript has been improved after revisions.